# "I was scared dating… who would take me with my status?"—Living with HIV in the era of UTT and U = U: A qualitative study in Johannesburg, South Africa

Tembeka Sineke[1], Dorina Onoya[1], Idah Mokhele[1], Refiloe Cele[1], Shubhi Sharma[2], Patience Sigasa[1], Mandisa Dukashe[3], Laila Hansrod[4], Robert Inglis[4], Rachel King[5], Jacob Bor[1,2,6]*

1 Faculty of Health Sciences, Department of Internal Medicine, Health Economics and Epidemiology Research Office, School of Clinical Medicine, University of the Witwatersrand, Johannesburg, South Africa, 2 Department of Global Health, Boston University School of Public Health, Boston, Massachusetts, United States of America, 3 HIV Survivors and Partners Network, Tshwane, South Africa, 4 Jive Media Africa, Pietermaritzburg, South Africa, 5 Institute for Global Health Sciences, University of California, San Francisco, California, United States of America, 6 Department of Epidemiology, Boston University School of Public Health, Boston, Massachusetts, United States of America

* jbor@bu.edu

## Abstract

South Africa rolled out Universal Test-and-Treat (UTT) in 2016, extending treatment eligibility to all persons living with HIV (PLHIV). We sought to understand how PLHIV in Johannesburg, South Africa, interpret and experience their HIV status, five years into the UTT era. In May 2021, we conducted in-depth interviews (IDI) (N = 27) with adult (≥18 years) PLHIV referred by HIV counsellors at three peri-urban primary healthcare clinics. We also conducted three focus group discussions (FGDs) (N = 27) with adult PLHIV recruited from clinics or from civil society organisations through snowball sampling. Follow-up interviews were conducted with 29 IDI and FGD participants. Participants were asked to reflect on their HIV diagnosis, what their HIV status meant to them and how, if at all, being HIV-positive affected their lives. Interviews and focus group discussions were audio-recorded, transcribed, translated to English, and analysed using a grounded theory approach. Participants perceived that HIV was common, that PLHIV could live a normal life with antiretroviral therapy (ART), and that ART was widely accessible. However, HIV elicited feelings of guilt and shame as a sexually transmitted disease. Participants used the language of "blame" in discussing HIV transmission, citing their own reckless behaviour or blaming their partner for infecting them. Participants feared transmitting HIV to others and felt responsible for avoiding transmission. To manage transmission anxiety, participants avoided sexual relationships, chose HIV-positive partners, and/or insisted on using condoms. Many participants feared–or had previously experienced–rejection by partners due to their HIV status and reported hiding their medication, avoiding disclosure, or avoiding relationships altogether. Most participants were not aware that undetectable HIV is untransmittable (U = U). Participants who were aware of U = U expressed less anxiety about transmitting HIV to others and greater confidence in having relationships. Despite perceiving HIV as a manageable chronic condition, PLHIV still faced

**Data Availability Statement:** All relevant data are within the paper and its Supporting Information files.

**Funding:** This study was funded by the National Institutes of Health (R34MH122323) awarded to JB and DO. The funders had no role in study design, data collection and analysis, decision to publish, or preparation of the manuscript.

**Competing interests:** The authors have declared that no competing interests exist.

transmission anxiety and fears of rejection by their partners. Disseminating information on U = U could reduce the psychosocial burdens of living with HIV, encourage open communication with partners, and remove barriers to HIV testing and treatment adherence.

## Introduction

How do people living with HIV (PLHIV) today interpret and experience their HIV status? HIV/AIDS was imbued with social meaning from its earliest days, as people sought to understand this new, mysterious, and lethal condition. In 1981, United States public health officials infamously linked AIDS to four already-stigmatized "risk groups"–homosexuals, heroin users, haemophiliacs, and Haitians. As Susan Sontag wrote, AIDS diagnosis became a metaphor for indulgence, deviance, and transgression, a stigma or a mark of personal failure [1]. In sub-Saharan Africa (SSA), different metaphors emerged based on epidemiological and cultural contexts. AIDS was linked to sex work, witchcraft, divine punishment for sinful acts, and attributed to immorality [2–7]. The negative social meaning associated with HIV was rooted in real fears of early death, leaving loved ones behind, and contagion. Stigma, however, has been profoundly injurious to the mental health of PLHIV and has discouraged HIV testing, status disclosure, and treatment and prevention uptake [8–14].

Antiretroviral therapy (ART) has transformed HIV/AIDS into a clinically-manageable chronic disease with limited impact on life expectancy [15] and economic productivity [16,17]. South Africa rolled out ART in the public sector in 2004, and approximately 5.1 million people were on therapy and virally suppressed in 2020 [18]. Access to ART is widespread, with 96% of South Africans living within 10km of a health facility providing ART [19]. As of September 2016, all PLHIV are eligible for ART regardless of CD4+ lymphocyte count, under South Africa's Universal Test-and-Treat (UTT) policy [20,21]. In 2020, 94% of South African PLHIV knew their HIV status, of which 74% were on treatment, of which 67% were virally suppressed [18]. With expanded access and eligibility for ART, HIV is no longer a death sentence. However, studies have found that HIV remains a stigmatized condition [22–24].

The goal of UTT–and an ongoing policy challenge of the UTT era–is to increase use of ART among PLHIV early in their infection, thereby reducing the risk of onward transmission. Although the efficacy of HIV treatment-as-prevention (TasP) was established in 2011, TasP has not historically been emphasized in HIV counselling in South Africa [25], resulting in low knowledge about the prevention benefits of ART [26]. Evidence from other contexts suggests that providing information on TasP, including the "Undetectable Equals Untransmittable (U = U)" message may alleviate stigma and improve self-image among PLHIV [27].

In this qualitative study, we sought to understand how PLHIV in Johannesburg, South Africa, experience and interpret their HIV status in the UTT era; how being HIV-positive affects their lives; and what role, if any, knowledge of TasP plays in their experience of living with HIV. Understanding the meaning ascribed to HIV and the challenges PLHIV face due to their status is critical to support psychosocial well-being and to motivate testing and ART adherence.

## Methods and materials

### Study design

This qualitative study was undertaken as formative research to guide the development and randomized evaluation of an intervention to integrate U = U/TasP into HIV counselling in South

Africa (NIH R34 Bor/Onoya) [28,29]. The larger project aims to determine whether informing patients about HIV treatment-as-prevention during HIV post-test and adherence counselling affects their knowledge and attitudes; stigma and wellbeing; as well as towards ART uptake and adherence. This formative research study consists of 27 in-depth interviews (IDI) and three focus group discussions (FGD) with PLHIV residing in and around Johannesburg, South Africa. Data was collected in May 2021.

### Data collection

Adults (≥18 years) living with HIV were recruited from peri-urban primary healthcare clinics in Johannesburg for participation in in-depth interviews. The study followed guidelines for the conduct and reporting of qualitative studies (S1 Text). Potential participants were identified and referred by lay HIV counsellors. Dedicated study staff with training and experience in qualitative interviews conducted all screening, consent procedures, and data collection. Potential participants were screened if they were well enough to provide written informed consent in the participant's preferred language. Interviews were conducted using an IDI guide (S2 Text), lasted approximately 45 minutes, and were conducted in a private space within the clinic.

Participant selection was guided by two goals. First, we sought to recruit a sample of adult PLHIV that were diverse in terms of gender, age, and duration on treatment, and who accessed care at public facilities. Second, we sought to recruit participants with different levels of knowledge about U = U, to better understand whether the knowledge of U = U shaped people's experience of living with HIV.

To achieve these goals, we started by recruiting PLHIV at a primary health clinic immediately after HIV diagnosis or counselling. Our recruitment focused on participants who were either ART naïve or had been on treatment for less than six months–to better capture the experiences of people coming to terms with their HIV status in the UTT era. Potential participants were referred by lay HIV counsellors and introduced to the study staff. Those who agreed to be part of the study participated in an in-depth interview, an appropriate method of data collection given privacy concerns for people who may still be accepting their HIV status. We excluded patients who were unable or too sick to participate, unwilling to provide consent, or planned to get treatment elsewhere. Additionally, women who were pregnant at HIV diagnosis were excluded because care processes differ for pregnant women.

In-depth interviews covered the following domains: 1) perceptions of and experiences with HIV diagnosis (including thoughts and feelings after diagnosis, experiences with disclosure, and impact on relationships with family, friends and sexual partners), 2) In-depth understanding of challenges PLHIV experienced related to their HIV status and 3) perceptions about TasP. Interviews were conducted in English, Sotho and Zulu.

All participants recruited from clinical settings were unaware of U = U, which we expected as U = U had not yet been integrated into HIV counselling. We therefore engaged in a second recruitment strategy, obtaining participants through a civil society organization that seeks to educate people on U = U. We used snowball sampling, starting with a few initial contacts, to recruit participants for three focus group discussions (FGDs), which we stratified by gender were conducted using an FGD guide (S3 Text). The FGDs allowed us to get multiple perspectives on the same theme and enabled us to have a discussion about U = U in which participants could hear other participants endorsing the science of U = U. These FGDs were stratified by gender to promote open dialogue among participants around topics involving sexual relationships. A third (n = 6) FGD consisted of younger (**Table 1**) participants.

**Table 1. Characteristics of patients included in the analysis.**

|  | IDIs | FGD1 | FGD2 | FGD3 |
|---|---|---|---|---|
|  | N = 27 (col%) | N = 9 (col%) | N = 12 (col%) | N = 6 (col%) |
| **Age at enrolment** |  |  |  |  |
| 18–30 | 24 (88.9) | N/A | N/A | 6 (100.0) |
| 30+ | 3 (11.1) | 9 (100.0) | 12(100.0) | N/A |
| **Gender** |  |  |  |  |
| Females | 23 (85.1) | 9 (100.0) | N/A | 6 (100.0) |
| Males | 4 (14.9) | N/A | 12(100.0) | N/A |
| **Recruitment source** |  |  |  |  |
| Civil society | N/A | 9 (100.0) | 12(100.0) | 6 (100.0) |
| Health facility | 27 (100.0) | N/A | N/A | N/A |

*We conducted follow-up interviews with a subset of 29 participants from both IDIs and FGDs to get a full understanding of their journey of living with HIV.

The FGDs were facilitated by trained study staff and lasted for approximately two hours, covering similar topics to in-depth interviews including, experiences with the HIV diagnosis, experiences with HIV treatment, perceptions about treatment-as-prevention, and questions around communication methods for U = U/TasP. The facilitator encouraged participation and ensured that every participant was given an opportunity to contribute to the discussions. A note taker carefully observed dynamics between the participants. The sessions were audio recorded and field notes were taken.

## Data analysis

Interviews and focus group discussions were audio-recorded, transcribed verbatim, and translated from both Sotho and Zulu to English. The original transcripts were compared against the translated by a member of research team to ensure validity. Each transcript was coded by two coders independently, and the coders held detailed discussions to reconcile differences in codes and interpretation. In cases where the difference could not be reconciled, a third member of the research team was assigned to help resolve the differences.

An inductive approach (S1 Table), as used in grounded theory, was used to derive emergent themes from our data. Since the themes were data driven, they did not mirror the exact questions asked of participants and rather sought to capture the "deep story(ies)" of our informants [30]. As emergent themes were identified and refined, new codes were developed, and transcripts revisited to identify subthemes and links between these constructs.

To maintain confidentiality and anonymity, all identifiers were removed from the final analytic data. This study was approved by the Human Research Ethics Committee (Medical) of the University of the Witwatersrand (M200529 MED20-05-019) and Boston University Medical Campus Institutional Review Board (H-40891).

## Results

### 1. Meaning ascribed to HIV and PLHIV self-image in the UTT era

The IDIs and FGDs yielded the following emergent themes. First, while PLHIV viewed HIV as a common chronic condition, they still perceived stigma associated with HIV, engaged the language of blame in talking about HIV transmission, and many had low self-image associated with being HIV-positive. Second, PLHIV experienced transmission anxiety, i.e. the fear of

passing HIV on to someone else; several reported fears of future rejection by romantic partners, disincentivizing disclosure, and many questioned whether they would be able to have sexual relationships or a family. Third, most participants recruited from the clinic were not aware, or were not confident, that ART leading to viral suppression prevents HIV transmission; however, those respondents who were aware of U = U/TasP had less anxiety about HIV transmission and had more positive self-image.

**1.1. Increased normalization of HIV.** HIV was seen as "normal" and "common" by most participants. And majority of the participants knew many people who were living with HIV. Treatment was ubiquitous, and HIV was no longer viewed as a death sentence. Many participants had friends and family members on ART, living healthy, normal lives:

*"I can't stop taking my medication because I am HIV positive, a lot of people are living normal lives, I'm not the only person and I'm not the first one."–**Female 23, IDI***

Direct experience of family and friends having success with ART gave people confidence that they too could live a normal life with HIV:

*"I have a younger brother who was born with the virus from my mother, he has been taking the treatment ever since. That is what gave me some motivation that if he could live for so long and be healthy then that means I can also do the same."–**Female 25, IDI***

Despite the ability to live a "normal" life with HIV, participants still reported feelings of guilt and shame related to their HIV diagnosis. In some cases, these feelings led to challenges in coping and accepting their HIV status and to delays in ART initiation:

*"To be honest I was never okay, I even started drinking a lot and I didn't take the treatment from September, October, November I think I only started taking the treatment in February this year."–**Female 26, IDI***

*"Umh like really, truly speaking; at first, I was really disappointed as I am not that type who is busy dating all the time. I have one partner but then I learned to accept because I have kids, I had to tell myself that I have to live for their sake."–**Female 27, IDI***

*"I didn't know how to feel. It was two different stories: I was pregnant, and I was HIV+. For me, I felt disappointed, but then, I was just thankful I was still alive."–**Female 26, IDI***

**1.2. Persistent HIV stigma is linked to sexual transmission.** Despite the waning association of HIV with death, HIV still remained highly stigmatised as a sexually transmitted disease. Stigma is a "mark of disgrace" and its link with transmission is twofold. First, HIV is stigmatized because it is linked to stigmatized sexual behaviour (e.g. promiscuity, infidelity) and thus inherits the stigma associated with that behaviour. Second, because HIV is stigmatized, sexual transmission of the virus involves giving an undesired "mark of disgrace" to an intimate partner. It is unsurprising, therefore, that many participants viewed HIV transmission to result from bad behavior or moral transgression. One participant cited "loss of discipline" as the reason for her HIV infection.

*"I started treatment in 2017 because I was in denial, was hiding it even at home, this thing, and I was scared to talk about this situation. . .. Unfortunately, at the time I broke my virginity, I got pregnant, and I got HIV and it was so sad to me that I had disciplined myself for such a long time."–**Female, FGD, civil society group***

Another respondent blamed herself for what she felt was reckless behaviour and felt a strong motivation to avoid onward transmission:

*"I took a risk, after taking that risk with my partner I saw him presenting HIV symptoms I knew it there and then that he is HIV positive because he had Shingles. So I don't want to infect someone else."*–**Female 26, IDI**

Participants described experiencing feelings of indignation at the carelessness of partners who did not try harder to protect them from HIV. They consequently felt a strong sense of responsibility to be careful and avoid transmitting HIV to others. Not having someone else to blame led to feelings of shame and forced people to reckon with their own role in transmission. Regardless of who was responsible in a given instance, a key theme was the use of language connoting culpability and blame.

*"The most embarrassing thing is not knowing who gave it to me. . . . If I had someone to blame. . . . But now I'm taking accountability. . .. I was careless and don't have anyone to blame."*–**Female 30, IDI**

Transmission was interpreted as a breach of trust within a relationship.

*"I didn't believe it at first because I only had one partner. I trusted that person and I didn't expect him to infect me with the virus. I was very confused because we have been together for years. . . . It took me a while to accept it. But as time went on, I ended up accepting."*–**Female 23, IDI**

## 2. Challenges PLHIV experienced related to their HIV status regarding relationships and disclosure

**2.1. Transmission anxiety.**  Although the stigma associated with HIV was linked to transmission, it had broader impacts on how PLHIV perceived themselves and shaped fears for their future. Participants voiced concerns such as "I don't know what will happen to me" "what the future will bring". Much of this sentiment derived from the way that HIV shaped people's experiences with current and future relationships, the topic to which we now turn.

Transmission-related HIV stigma had a profound impact on PLHIV's expectations and experiences of sexual relationships. Participants feared transmitting HIV to their sexual partners, felt responsible for avoiding transmission, and feared being stigmatised by others for being a risk of transmitting HIV to other people. Both men and women feared transmitting HIV to others and that fear weighed on them:

*"The only thing that scares me is the possibility of transmitting this virus."*–**Female, FGD, civil society group**

*"I can't sleep with someone without a condom" "I don't want to infect another person with this thing" "I am scared [of infecting someone]."* **Male 53, IDI**

Some participants avoided relationships altogether, others chose future partners based on their HIV status, and others insisted on condom use although this was not always consistent. Some participants avoided relationships, delayed sex, or chose HIV-positive partners to manage anxiety around transmitting HIV to others.

*"But for a couple of months, I think I had this anxiety; I believe I still have it even now because I would meet someone and then as the relationship progresses, I would always run away from the sex talk, whether he is HIV positive or negative. I've tried dating someone who's HIV positive, I met him in one of those support groups you get on Facebook, and with him also I was still not comfortable, . . . I used to fear that we would be busy and then he will take it out also, and I don't know what he has; and you know people worry about HIV only and I'm thinking "I can't have two or three STD's", so yeah, it's something I'm still working on, the anxiety! It's still there."*–**Female 30, IDI**

Dating other PLHIV was one way participants sought to manage transmission anxiety:

*"Well, I think I still fear infecting someone else because I still use HIV as a criteria, like a searching criteria so when I look at my list I'm like 'Okay God at least he must be HIV positive also.' I have blocked the idea of dating someone who's HIV negative in my head, because I felt that it's going to be uncomfortable for them whenever I have to take the pill in from of them. If I tell him that 'this week I'm going through pill fatigue' they wouldn't understand, instead they would think 'You are being selfish, take your pill, you must protect me also.'"*–**Female 30, IDI**

One female PLHIV even rejected an HIV-negative partner because she didn't feel comfortable being intimate with him and didn't want to "waste his time".

*"I remember this one guy who came to my workplace, we met, we had a conversation and he asked me out. Two weeks later, because I felt 'I'm falling for this person', I used my [HIV] status to now push him away, and he just said, 'So what?' But we broke up because I wasn't ready to get intimate, I just told him 'I don't want to waste your time, I don't think I'll be able to be intimate with you anytime soon so I'd rather set you free now'; at first, he could understand, I said no. let me deal with myself first and then I will try again but for now NO."*–**Female 30, IDI**

Some participants used condoms to avoid transmission:

*"Yes it (relationship) changed. I now realised that I always have to use a condom whenever I engage in sex."*–**Female 21, IDI**

However, condom use was inconsistent, as some participants had partners who did not want to use condoms.

*"He refused to use a condom first of all. Obviously that led to me falling pregnant. Before falling pregnant, I experienced a lot of STI's. One of them being HPV, Syphilis, Gonorrhoea, I had a whole list of them, that's what took me to the clinic at the end of the day, at the beginning, before I even fell pregnant. So, I was in and out of the clinic, my parents didn't even know that I went to the clinic."*–**Female, FGD, civil society group**

Others recognized that condoms could not be used if partners were trying to conceive.

*"When it comes to my future partner, we will use protection, but when the time comes then he decides we should have a baby, I would then ask him to come with me to the clinic. He will receive counselling, and maybe they will tell him how and what to do."* **Female 30, IDI**

**2.2. Reputation loss, fear of rejection, and challenges finding a partner.** Participants expected judgment and rejection from potential partners, friends and their families, and worried that peers would gossip about their status. These fear and negative experiences added to disclosure challenges. Some participants chose not to disclose their status manage fear of rejection, while others saw early disclosure as imperative to identifying and protecting a supportive partner. Disclosure fears also affected ART adherence, as some people felt they had to hide their medication or not take their medication if they were going out with friends. Some PLHIV feared that their HIV status would be weaponized against them via gossip or perceived shaming.

*"If sometimes I fight with someone, maybe a relative. They tell you about your status [reveal my status to others], that's why I don't want to tell them."–**Male 53, IDI***

Participants bristled at the loss of privacy if their HIV status became known to their peers.

*"Some were supportive though some were judgemental, like one of my friends would just ask me randomly 'Hey Sisi do you take your treatment?' I would just respond and say 'You are not going to tell me that I must take my treatment because I know my status.'"–**Female 26, IDI***

Many participants worried that their HIV status would reduce their ability to find relationship partners. These concerns were often based on direct experience including past rejections.

*"And I was scared dating. . . partners rejected me. . . who would take me [as a partner] with my status."–**Female 30**, IDI*

And fear of rejection led some PLHIV to think twice about entering a relationship.

*"Right now there is this guy who is showing interest in me. I was still thinking about, if I agree to start a relationship with him, will he understand my situation? I will have to explain my situation to him. I don't know if he will accept it or will just carry on with his life."–**Female, FGD, civil society group***

Pessimism about finding a supportive partner was associated with a negative future outlook. Some PLHIV saw key life milestones–having a long-term relationship with a desirable partner, having children–as out of reach because of their status.

*"I don't want to lie, I am scared because I'm still very young and I don't know what life has for me, and what kind of partner am I going to find, is he going to accept it or not and also I don't have a child so I don't know, I don't want to lie I really do not know. However, that will never stop me from taking my treatment."–**Female 26, IDI***

*"At the time I thought I wouldn't be able to have kids."–**Female 30, IDI***

**2.3. Disclosure challenges.** Disclosing one's HIV status can be a difficult conversation, and some participants–particularly younger participants sought to avoid it. For example, one female participant described "disclosing", using an indirect method, by taking her medication in front of her partner, without actually talking about her HIV status or how she and her partner would navigate it:

*"[Our relationship] has changed because we did not talk for quite some time. He stays in Durban and at the time I was in the Eastern Cape. We did not talk and he did not want me to visit him. He then decided around December that I can visit him. We never spoke about it, I used to take my treatment in front of him, I don't know if he would hide his or what, but I would take mine at the time when I had to. We are okay now, we communicate."–Female 26, IDI*

While another male previously experienced a partner leaving the relationship after disclosure:

*"There is no way to tell if I were to tell her by day two if she would have left. I told her by day 19 and she was already gone. So, HIV on its own still have some issues when it comes to relationships. Quite a lot, I just don't know, how can we, but I don't blame them. I looked back and say if I really didn't do this, I just go, with anyone who have it, I don't think I would. So, I do not want to put the burden on people who are HIV negative, to say come love me as I am. It is impossible."–Male, FGD, civil society group*

A young male participant describes a similar scenario in which disclosure occurred indirectly rather than through direct communication. Due to its stigma, HIV remains difficult to talk about within a relationship:

*"In my past relationship, it happened that my girlfriend was aware that this person is going through this, she saw my tablets but didn't have the guts to ask me. Her friend found a way to ask me and I openly told her friend. After that I realised she was distant."–Male 21, IDI*

Older participants, who were likely more relationship-experienced, saw early disclosure as a critical "screening tool" to select a partner that would support them regardless of their HIV status:

*"On the first day I meet her I tell her I am HIV positive; it will be up to her whether she continues, or she leaves."–Male 53, IDI*

## 3. Understanding of U = U/TasP and its role in self-image and relationships

Our study respondents were recruited from two populations: PLHIV referred from public sector clinics and PLHIV identified through a civil society organization that works on TasP/U = U. Most participants recruited from the clinic were not aware, or were not confident, that ART leading to viral suppression prevents HIV transmission. For example one participant recruited from the clinic reported:

*"Yes I've heard that on Facebook but they said after some time when taking your treatment you can't transmit. But, I don't know how true that is."–Female 26, IDI*

PLHIV who knew about U = U/TasP and had confidence in the science described the feeling of knowing that they would not infect their partner so long as they stayed on their medication. The quotes convey a significant reduction in anxiety, the sense of a weight being lifted, and the way that U = U/TasP opened up opportunities to live a normal life.

*"If you stick to your treatment you don't have to worry about infecting your partner. [Being virally suppressed,] I feel normal. I feel like anyone else. I can start a relationship with anyone I want."–Female 30, civil society group*

*"I was so much looking forward to the day when I would find out I was virally suppressed; my goal was to see the VL going down; when it was <40 copies, I knew I couldn't infect someone else. Now I didn't have to worry about HIV. . .. You see a very bright future". [respondent was privy to the science of U = U early because he was in HIV research]*–**Male, FGD, civil society group**

*"I'm not worried about transmitting to someone else because I know that when I'm virally suppressed, I can't transmit, and I am virally suppressed."*–**Male, civil society group**

Having information on TasP/U = U also armed PLHIV with the confidence to have more open conversations about HIV within their relationship.

*"I disclosed my status to my baby daddy, and then. . . he was scared and it was heavy, 'cause you know, men are weak. I'm sorry to say that. But men are weak. To understand that they have fear over a lot of things. He was very much concerned about the child again. But I got information, I educated him about U = U, and how we are going to save the child from getting infected by the virus. I was taking my treatment, I was using the condom when having sex. That was the only thing that saved my child from getting HIV"*–**Female 19, IDI**

Finally, the ability to control transmission risk through daily ART adherence led to a reduction in internalized stigma and the sense that PLHIV could be fully moral actors in the world.

*"Being virally suppressed means I'm a full human again; HIV is no longer an issue; what is for me is to keep it suppressed;. . .I didn't feel comfortable shaking hands initially, but now. . .I can really stand up and just be myself; I can talk about HIV without feeling like I'm a victim."*–**Male, FGD, civil society group**

## Discussion

This paper aimed to look at how do PLHIV experience their HIV status in the UTT era and whether the meaning of HIV evolved as the epidemic matured. We conducted interviews and focus groups with PLHIV in Johannesburg, South Africa, in May 2021 to better understand contemporary experiences of people living with the virus.

The scale-up of ART has transformed HIV into a manageable chronic condition. Universal eligibility under UTT means that PLHIV do not have to wait to start therapy and that PLHIV have access to ART as a means to prevent onwards transmission. Still, while participants in our study viewed HIV as "normal" and "common", we found that internalised stigma remained pervasive. Our findings echo those of other studies that have found that stigma, though reduced [31], remains high despite the scale-up of treatment [32,33]. Similarly, recent findings showed little evidence that UTT substantially reduced HIV stigma among PLHIV, community members, and health workers (HCWs) not living with HIV [34]. Our results are particularly concerning as studies have shown that stigma is associated with depression among PLHIV and negatively impacts quality of life [35,36]. Additionally, stigma is a barrier to HIV testing and ART adherence, and, thus undermines efforts towards the elimination of HIV [37,38]. HIV stigma is often internalized, leading to continued psychosocial harm [6]. Our findings show that study respondents reported feelings of shame and guilt for contracting the virus and perceived their infection as a moral transgression. These findings are consistent with prior research in sub-Saharan Africa, which showed that HIV infection was perceived to be a result of risky sexual behaviour [2,7].

Alongside the shame of contracting HIV, respondents felt a deep responsibility to avoid transmitting the virus to others. Respondents reported fear and anxiety around transmission as a major concern. Prior literature suggests that PLHIV are deeply concerned about transmission and engage in a variety of behaviours to protect partners from infection, a phenomenon known as "HIV prevention altruism" [39]. To manage transmission anxiety, participants avoided relationships, avoided sex, choose to date other PLHIV, or insisted on condom use. These results are consistent with previous findings whereby individuals reported diminished sexual desire due to fears of transmitting HIV to others [40,41]. Engaging in preventive behaviors to limit transmission represented an important zone of "moral agency" for PLHIV. For those who knew about TasP/U = U, the ability to effectively prevent transmission through daily ART adherence was tremendously freeing and an important burden lifted.

Another major concern was fear of rejection, which manifested in disclosure challenges. Some participants chose to conceal their status due to fear of judgement and stigma from their family, friends and/or partners. In some instances, indirect disclosure of HIV status occurred despite participants efforts to conceal status. Non-disclosure due to stigma can result in additional challenges for PLHIV including interruptions in treatment and can negatively impact quality of life [42]. Conversely, several studies show that individuals who disclose their status are more likely to receive support from their loved ones and are more likely to adhere to treatment [43]. Fear of rejection was also linked to internalized stigma, with several participants conveying that partners would be well within their rights to reject them because they were HIV-positive, implying that they perceived themselves to be a less worthy partner than someone who was HIV-negative.

Most participants in our study did not have knowledge of TasP/U = U and this is consistent with recent empirical work from South Africa [26] and a global systematic review [27]. In our study, the only participants that were knowledgeable about TasP were those recruited from a civil society organization that does advocacy work on TasP/U = U. These participants expressed that they were no longer worried about infecting their partners. Moreover, they felt confident disclosing their status to others because they knew that they could not infect someone else. And they no longer feared rejection because their partners had no good reason to reject them.

Our findings underscore the significant extent to which messaging on TasP/U = U may address the significant hardships experienced by PLHIV due to persistent transmission-related stigma, anxiety, and shame. It is therefore paramount to ensure that U = U/TasP information is disseminated widely [44] as it offers an opportunity to reduce stigma and discrimination against PLHIV [45]. The message of U = U/TasP offers people the opportunity to exercise their moral agency, to have full lives with families, sex, and relationships without feeling like they are a threat to others. Equipping PLHIV and the society at large with sufficient knowledge is a powerful tool to eliminate misinformation and break the cycle of stigma.

The results should be considered in light of several study limitations. First, the study population is not representative of the general population. Our study was conducted in a predominantly peri-urban urban setting with only a few health facilities included. The study is subject to potential bias due to some of the participants recruited via referrals. In particular, the participants who were knowledgeable about TasP/U = U were identified through a U = U advocacy organization and may have had greater confidence in TasP and greater acceptance of their HIV status than other persons knowledgeable of the science. Similarly, the participants recruited for IDIs were in care and knowledgeable of benefits of ART through experience and during counselling. Responses might have been different if the participants were not engaged in clinical care and recruited from clinical settings. We also did not explicitly recruit groups of people that are at higher risk for HIV acquisition including members of the LGBTQIA

+ population, sex workers, and people who inject drugs. Despite these limitations, we were able recruit a study population via purposive sampling that was diverse in terms of gender, age and level of U = U/TasP knowledge. We invited participants who were willing to share their journeys openly. Second, although we reached saturation in our key themes, the sample size was limited, and this study may need to be replicated in other settings and augmented by quantitative surveys to further establish generalizability. Third, majority (70.4%) of our participants were female. Despite conducting an all-male focus group, a limitation of the study is the small number of interviews conducted with recently diagnosed, younger male respondents. Results may not be generalizable to this population. Fourth, coding/analysis of qualitative data is subjective. To mitigate the impact of potential rater bias, we had at least two coders (TS, RC, JB) code all transcripts and held detailed discussions to reconcile differences in codes and interpretation.

## Conclusion

Persistent stigma in the UTT era is rooted in the sexual transmission of HIV. PLHIV experience self-stigma related to their status, experience anxiety about transmitting HIV to others, and fear rejection from partners based on their HIV status. Our findings suggest that expanding ART eligibility alone is not enough to address the challenges. Disseminating information on TasP/U = U could reduce the psychosocial burdens of HIV including self/internalized stigma, encourage disclosure, and remove barriers to HIV testing and treatment adherence.

## Supporting information

**S1 Table. Codebook for qualitative analysis.**
(PDF)

**S1 Text. Consolidated criteria for reporting qualitative research (COREQ) checklist.**
(PDF)

**S2 Text. In-depth interview (IDI) guide.**
(PDF)

**S3 Text. Focus group discussion (FGD) guide.**
(PDF)

## Acknowledgments

The authors would like to thank the members of HIV Survivors and Partners Network for assistance in identifying potential participants. We also acknowledge the dedication of Ms Smangele (Patience) Sigasa, Ms Hazel Tau, and Ms Sharon Kgowedi in conducting the interviews and discussions.

## Author Contributions

**Conceptualization:** Tembeka Sineke, Dorina Onoya, Jacob Bor.

**Data curation:** Jacob Bor.

**Formal analysis:** Tembeka Sineke, Dorina Onoya, Jacob Bor.

**Funding acquisition:** Dorina Onoya, Jacob Bor.

**Investigation:** Tembeka Sineke, Dorina Onoya, Jacob Bor.

**Methodology:** Tembeka Sineke, Dorina Onoya, Jacob Bor.

**Project administration:** Tembeka Sineke, Patience Sigasa.

**Supervision:** Dorina Onoya.

**Validation:** Tembeka Sineke, Dorina Onoya, Jacob Bor.

**Visualization:** Dorina Onoya, Jacob Bor.

**Writing – original draft:** Tembeka Sineke.

**Writing – review & editing:** Tembeka Sineke, Dorina Onoya, Idah Mokhele, Refiloe Cele, Shubhi Sharma, Patience Sigasa, Mandisa Dukashe, Laila Hansrod, Robert Inglis, Rachel King, Jacob Bor.

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
