## [Decision Letter · Decision Letter 0]

9 Aug 2022

PGPH-D-22-01077

“I was scared dating… who would take me with my status?”- Living with HIV in the UTT era in Johannesburg, South Africa

Dear Dr. Sineke,

Thank you for submitting your manuscript to PLOS Global Public Health. After careful consideration, we feel that it has merit but does not fully meet PLOS Global Public Health’s publication criteria as it currently stands. Therefore, we invite you to submit a revised version of the manuscript that addresses the points raised during the review process.

We look forward to receiving your revised manuscript.

Kind regards,

Sarah Brewer, PhD

Academic Editor

Journal Requirements:

1. In the online submission form, you indicated that "The authors declare that the data will be made available.". All PLOS journals now require all data underlying the findings described in their manuscript to be freely available to other researchers, either 1. In a public repository, 2. Within the manuscript itself, or 3. Uploaded as supplementary information.

Additional Editor Comments (if provided):

This manuscript describes a qualitative study using interviews and focus groups to understand experiences of stigma about HIV in an era of UTT and TasP in South Africa. There are important findings regarding the experiences of UTT and TasP herein. However, there are a number of reviewer concerns about reporting of methods and findings that need to be resolved. Please pay attention especially to Reviewers 2 and 3 requests for additional information about methods and details of the findings in your revisions. As Reviewer 2 suggests, the COREQ checklist will be helpful for the authors to ensure important methodological and analytic process aspects are reported. Authors also need to address any differences in findings between patients recruited to interviews vs. community at-large recruited to the focus groups as well as limitations due to the overwhelmingly female sample. Reviewers suggest a number of other revisions to clarify your meaning and strengthen the manuscript prior to publication.

Reviewers' comments:

Reviewer's Responses to Questions

**Comments to the Author**

1. Does this manuscript meet PLOS Global Public Health’s publication criteria? Is the manuscript technically sound, and do the data support the conclusions? The manuscript must describe methodologically and ethically rigorous research with conclusions that are appropriately drawn based on the data presented.

Reviewer #1: Yes

Reviewer #2: Partly

Reviewer #3: Partly

2. Has the statistical analysis been performed appropriately and rigorously?

Reviewer #1: N/A

Reviewer #2: N/A

Reviewer #3: N/A

3. Have the authors made all data underlying the findings in their manuscript fully available (please refer to the Data Availability Statement at the start of the manuscript PDF file)?

Reviewer #1: Yes

Reviewer #2: Yes

Reviewer #3: No

4. Is the manuscript presented in an intelligible fashion and written in standard English?

Reviewer #1: Yes

Reviewer #2: Yes

Reviewer #3: Yes

5. Review Comments to the Author

Reviewer #1: 1. The IDIs and FGDs had mutually exclusive participants or there was an overlap? Please clarify.

2. Since this is framed around UTT, can authors include how experiences changed before and after UTT? I am not able to see the utility of UTT otherwise.

3. “Despite the normalization of HIV as a chronic disease” is very loaded since normalization can mean many different things. May consider changing it and making it more specific.

4. ‘Participants did not view HIV as a death sentence with treatment widely available’ – this is repetition of previous line in different words. May consider removing since it doesn’t add any value.

5. The paragraph following ‘1.2. Persistent HIV stigma is linked to sexual transmission’ doesn’t have material related to this specific heading.

Reviewer #2: 1. The numbers do not add up in the result section of the abstract. Kindly review.

2. Don't you think the clinic setting had an influence on the responses for the in-depth interviews?

3. Although this study is said to be part of a larger one which has its objectives, how were the domains/ themes objectively identified/ determined for the qualitative study? what was the methodological orientation or theory

4. Since you reported purposive sampling( though not very clearly done), what guided participant selection? Did you consider duration on treatment in selecting potential respondents, since this has been shown to influence stigma-related responses and client knowledge of their condition.

5. How objective was the data analysis? sharing the coding tree, as prescribed by Consolidated criteria for reporting qualitative research (COREQ) would be helpful.

6. Who did the interviewing? what was their relationship with the respondents? what is their qualification?

7. The quotation between 57 and 62 is not very clear. Could you kindly look at it again?

8. The quotation in 134-136 don't seem to support the statement in 131. Kindly clarify

9. Did clients validate the findings?

10 Can you provide information on those who refused to take part in the study?

11. Although you said you tried to balance the genders in selecting respondents, the quotations were largely from females. Does this not affect the rigor of your findings?

12. Table 1 shows that you had 21 participants for the first FGD. How were they managed to ensure that contributions were received from all participants. How was data collected during the focus group discussions? was it also video taped or were field notes taken?

13. The text mentions a third FGD but table 1 has only two. Kindly look at that again.

14. How was the data collected in Sotho and Zulu languages managed well before analysis?

Reviewer #3: This manuscript presents qualitative data concerning PLWHA in South Africa's experiences with stigma and relationships in the era of UTT and TasP. The manuscript presents interesting findings, with potential implications for patient counselling and support, but there are several areas where the manuscript could be strengthened.

Minor comments:

The data availability statement is a little vague, and it isn’t clear to me whether it is feasible or ethical to make the qualitative datasets available, without explicit prior permission from the participants.

I would check the keywords and spell out some of them (especially Treatment as Prevention rather than TasP), and consider adding “qualitative methods” as a keyword.

Introduction:

- Would include where South Africa is at with their 90-90-90/95-95-95 goals

- The introduction covers a lot of ground, but it isn’t clear what gaps this study intends to fill. It seems like the goal of the study is to understand how UTT and TasP impacts stigma and people’s lives(?), but this needs to be more clearly stated, and additional literature provided. The PopART study may be a useful place to look, as they’ve done a fair amount of work on stigma in the context of promoting TasP.

Methods

- The rationale for doing both in-depth interviews and FGDs is unclear, especially since the topics covered seem to be the same?

- The sampling frame and inclusion criteria for the in-depth interviews is also rather vague.

- The thematic analysis approach needs to be a little clearer, and it would be helpful if the authors provided citations to support their analytical approach. The coding process is also missing some details, including how many coders there were and their backgrounds/positionality. Re-working the methods in line with the COREQ checklist would help clarify a lot of the questions.

- Including the interview and FGD guides, as well as the final codebook as supplemental materials would be helpful.

Results

- The results would benefit from an introductory paragraph mapping out the themes and sub-themes to be described.

- The ID numbers attached to the quotes are not necessary, since they do not ,mean ,much to the reader. I would either drop or replace with something more meaningful (e.g., demographic information like age, time since diagnosis, other).

- I think making the manuscript objectives and guiding framework(s) clearer in the introduction would make the organization and flow of the results a little smoother.

Discussion

- The lack of male participants in the qualitative sample also seems like a potential limitation. I’m curious as to why there wasn’t more of an effort to balance the sample, but maybe this wasn’t important for the research questions? Would be interesting to know the extent to which participants concerns are gendered, especially given all the concern women expressed about future reproductive health, pregnancies.

- Given how large the literature on HIV stigma is, and the growing literature on UTT and TasP, the discussion could do a little more in term of linking findings to this larger literature, as well as making recommendations.

6. PLOS authors have the option to publish the peer review history of their article (what does this mean?). If published, this will include your full peer review and any attached files.

**Do you want your identity to be public for this peer review?** For information about this choice, including consent withdrawal, please see our Privacy Policy.

Reviewer #1: No

Reviewer #2: No

Reviewer #3: No

---

## [Decision Letter · Decision Letter 1]

18 Aug 2023

“I was scared dating… who would take me with my status?”- Living with HIV in the era of UTT and U=U: a qualitative study in Johannesburg, South Africa

PGPH-D-22-01077R1

Dear Dr. Bor,

We are pleased to inform you that your manuscript '“I was scared dating… who would take me with my status?”- Living with HIV in the era of UTT and U=U: a qualitative study in Johannesburg, South Africa' has been provisionally accepted for publication in PLOS Global Public Health.

Best regards,

Sarah E. Brewer, PhD

Academic Editor

Please note that while the reviewers recommend acceptance, and I agree as editor, I still recommend that you ensure your data accessibility statement is clear and consistent before publication.

Reviewer Comments (if any, and for reference):

Reviewer's Responses to Questions

**Comments to the Author**

1. If the authors have adequately addressed your comments raised in a previous round of review and you feel that this manuscript is now acceptable for publication, you may indicate that here to bypass the “Comments to the Author” section, enter your conflict of interest statement in the “Confidential to Editor” section, and submit your "Accept" recommendation.

Reviewer #1: All comments have been addressed

Reviewer #2: All comments have been addressed

Reviewer #3: All comments have been addressed

2. Does this manuscript meet PLOS Global Public Health’s publication criteria? Is the manuscript technically sound, and do the data support the conclusions? The manuscript must describe methodologically and ethically rigorous research with conclusions that are appropriately drawn based on the data presented.

Reviewer #1: Yes

Reviewer #2: Yes

Reviewer #3: Yes

3. Has the statistical analysis been performed appropriately and rigorously?

Reviewer #1: N/A

Reviewer #2: N/A

Reviewer #3: N/A

4. Have the authors made all data underlying the findings in their manuscript fully available (please refer to the Data Availability Statement at the start of the manuscript PDF file)?

Reviewer #1: Yes

Reviewer #2: Yes

Reviewer #3: No

5. Is the manuscript presented in an intelligible fashion and written in standard English?

Reviewer #1: Yes

Reviewer #2: Yes

Reviewer #3: Yes

6. Review Comments to the Author

Reviewer #1: Greetings, thank you for submitting the revised manuscript and providing a response to the reviewers. I note that you have carefully addressed all the points raised. While the manuscript can use some more contextualization around UTT, I can not say the manuscript in current state can be declined since it is a sound submission.

Reviewer #2: (No Response)

Reviewer #3: I appreciate the authors' responsiveness to the reviewers' comments. I think the comments have been adequately addressed. One minor comment--the data availability statement in the manuscript doesn't quite match what the authors wrote in their response to reviewers.

7. PLOS authors have the option to publish the peer review history of their article (what does this mean?). If published, this will include your full peer review and any attached files.

**Do you want your identity to be public for this peer review?** For information about this choice, including consent withdrawal, please see our Privacy Policy.

Reviewer #1: No

Reviewer #2: No

Reviewer #3: No
